# Unsupervised Neural Machine Translation

**Mikel Artetxe, Gorka Labaka & Eneko Agirre**
IXA NLP Group
University of the Basque Country (UPV/EHU)
{mikel.artetxe,gorka.labaka,e.agirre}@ehu.eus

**Kyunghyun Cho**
New York University
CIFAR Azrieli Global Scholar
kyunghyun.cho@nyu.edu

## Abstract

In spite of the recent success of neural machine translation (NMT) in standard benchmarks, the lack of large parallel corpora poses a major practical problem for many language pairs. There have been several proposals to alleviate this issue with, for instance, triangulation and semi-supervised learning techniques, but they still require a strong cross-lingual signal. In this work, we completely remove the need of parallel data and propose a novel method to train an NMT system in a completely unsupervised manner, relying on nothing but monolingual corpora. Our model builds upon the recent work on unsupervised embedding mappings, and consists of a slightly modified attentional encoder-decoder model that can be trained on monolingual corpora alone using a combination of denoising and backtranslation. Despite the simplicity of the approach, our system obtains 15.56 and 10.21 BLEU points in WMT 2014 French $\rightarrow$ English and German $\rightarrow$ English translation. The model can also profit from small parallel corpora, and attains 21.81 and 15.24 points when combined with 100,000 parallel sentences, respectively. Our implementation is released as an open source project[1].

## 1 Introduction

Neural machine translation (NMT) has recently become the dominant paradigm to machine translation (Bahdanau et al., 2014; Sutskever et al., 2014). As opposed to the traditional statistical machine translation (SMT), NMT systems are trained end-to-end, take advantage of continuous representations that greatly alleviate the sparsity problem, and make use of much larger contexts, thus mitigating the locality problem. Thanks to this, NMT has been reported to significantly improve over SMT both in automatic metrics and human evaluation (Wu et al., 2016).

Nevertheless, for the same reasons described above, NMT requires a large parallel corpus to be effective, and is known to fail when the training data is not big enough (Koehn & Knowles, 2017). Unfortunately, the lack of large parallel corpora is a practical problem for the vast majority of language pairs, including low-resource languages (e.g. Basque) as well as many combinations of major languages (e.g. German-Russian). Several authors have recently tried to address this problem using pivoting or triangulation techniques (Chen et al., 2017) as well as semi-supervised approaches (He et al., 2016), but these methods still require a strong cross-lingual signal.

In this work, we eliminate the need of cross-lingual information and propose a novel method to train NMT systems in a completely unsupervised manner, relying solely on monolingual corpora. Our approach builds upon the recent work on unsupervised cross-lingual embeddings (Artetxe et al., 2017; Zhang et al., 2017). Thanks to a shared encoder for both translation directions that uses these fixed cross-lingual embeddings, the entire system can be trained, with monolingual data, to reconstruct its input. In order to learn useful structural information, noise in the form of random token swaps is introduced in this input. In addition to denoising, we also incorporate backtranslation

---

[1]https://github.com/artetxem/undreamt

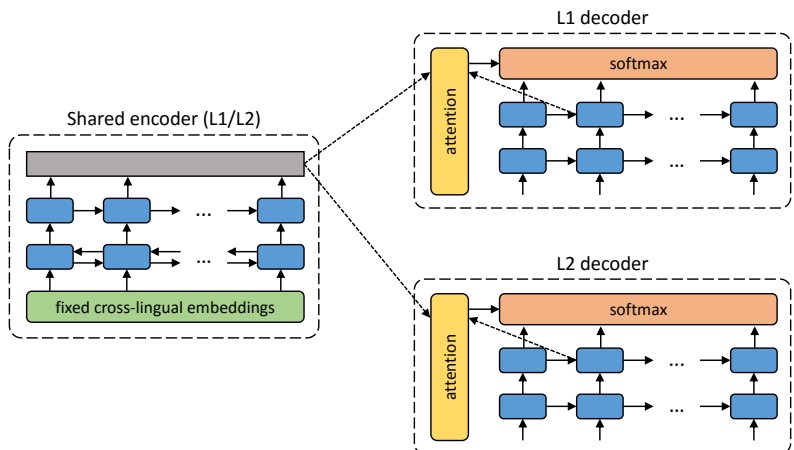

Figure 1: Architecture of the proposed system. For each sentence in language L1, the system is trained alternating two steps: *denoising*, which optimizes the probability of encoding a noised version of the sentence with the shared encoder and reconstructing it with the L1 decoder, and *on-the-fly backtranslation*, which translates the sentence in inference mode (encoding it with the shared encoder and decoding it with the L2 decoder) and then optimizes the probability of encoding this translated sentence with the shared encoder and recovering the original sentence with the L1 decoder. Training alternates between sentences in L1 and L2, with analogous steps for the latter.

(Sennrich et al., 2016a) into the training procedure to further improve results. Figure 1 summarizes this general schema of the proposed system.

In spite of the simplicity of the approach, our experiments show that the proposed system can reach up to 15.56 BLEU points for French → English and 10.21 BLEU points for German → English in the standard WMT 2014 translation task using nothing but monolingual training data. Moreover, we show that combining this method with a small parallel corpus can further improve the results, obtaining 21.81 and 15.24 BLEU points with 100,000 parallel sentences, respectively. Our manual analysis confirms the effectiveness of the proposed approach, revealing that the system is learning non-trivial translation relations that go beyond a word-by-word substitution.

The remaining of this paper is organized as follows. Section 2 analyzes the related work. Section 3 then describes the proposed method. The experimental settings are discussed in Section 4, while Section 5 presents and discusses the obtained results. Section 6 concludes the paper.

## 2  RELATED WORK

We will first discuss unsupervised cross-lingual embeddings, which are the basis of our proposal, in Section 2.1. Section 2.2 then addresses statistical decipherment, an SMT-inspired approach to build a machine translation system in an unsupervised manner. Finally, Section 2.3 presents previous work on training NMT systems in different low-resource scenarios.

### 2.1  UNSUPERVISED CROSS-LINGUAL EMBEDDINGS

Most methods for learning cross-lingual word embeddings rely on some bilingual signal at the document level, typically in the form of parallel corpora (Gouws et al., 2015; Luong et al., 2015a). Closer to our scenario, embedding mapping methods independently train the embeddings in different languages using monolingual corpora, and then learn a linear transformation that maps them to a shared space based on a bilingual dictionary (Mikolov et al., 2013a; Lazaridou et al., 2015; Artetxe et al., 2016; Smith et al., 2017). While the dictionary used in these earlier work typically contains a few thousands entries, Artetxe et al. (2017) propose a simple self-learning extension that gives comparable results with an automatically generated list of numerals, which is used as a shortcut for

practical unsupervised learning. Alternatively, adversarial training has also been proposed to learn such mappings in an unsupervised manner (Miceli Barone, 2016; Zhang et al., 2017).

## 2.2 STATISTICAL DECIPHERMENT FOR MACHINE TRANSLATION

There is a considerable body of work in statistical decipherment techniques to induce a machine translation model from monolingual data, which follows the same noisy-channel model used by SMT (Ravi & Knight, 2011; Dou & Knight, 2012). More concretely, they treat the source language as ciphertext, and model the process by which this ciphertext is generated as a two-stage process involving the generation of the original English sequence and the probabilistic replacement of the words in it. The English generative process is modeled using a standard n-gram language model, and the channel model parameters are estimated using either expectation maximization or Bayesian inference. This approach was shown to benefit from the incorporation of syntactic knowledge of the languages involved (Dou & Knight, 2013; Dou et al., 2015). More in line with our proposal, the use of word embeddings has also been shown to bring significant improvements in statistical decipherment for machine translation (Dou et al., 2015).

## 2.3 LOW-RESOURCE NEURAL MACHINE TRANSLATION

There have been several proposals to exploit resources other than direct parallel corpora to train NMT systems. The scenario that is most often considered is one where two languages have little or no parallel data between them but are well connected through a third language (e.g. there might be little direct resources for German-Russian but plenty for German-English and English-Russian). The most basic approach in this scenario is to independently translate from the source language to the pivot language and from the pivot language to the target language. It has however been shown that the use of more advanced models like a teacher-student framework can bring considerable improvements over this basic baseline (Firat et al., 2016b; Chen et al., 2017). In the same line, Johnson et al. (2017) show that a multilingual extension of a standard NMT architecture performs reasonably well even for language pairs for which no direct data was given during training.

In addition to that, there have been several attempts to exploit monolingual corpora for NMT in combination with the more scarce parallel corpora. A simple yet effective approach is to create a synthetic parallel corpus by backtranslating a monolingual corpus in the target language (Sennrich et al., 2016a). At the same time, Currey et al. (2017) showed that training an NMT system to directly copy target language text is also helpful and complementary with backtranslation. Finally, Ramachandran et al. (2017) pre-train the encoder and the decoder in language modeling.

To the best of our knowledge, the more ambitious scenario where an NMT model is trained from monolingual corpora alone has never been explored to date, but He et al. (2016) made an important contribution in this direction. More concretely, their method trains two agents to translate in opposite directions (e.g. French $\rightarrow$ English and English $\rightarrow$ French), and make them teach each other through a reinforcement learning process. While promising, this approach still requires a parallel corpus of a considerable size for a warm start (1.2 million sentences in the reported experiments), whereas our work does not use any parallel data at all.

## 3 PROPOSED METHOD

This section describes the proposed unsupervised NMT approach. Section 3.1 first presents the architecture of the proposed system, and Section 3.2 then describes the method to train it in an unsupervised manner.

## 3.1 SYSTEM ARCHITECTURE

As shown in Figure 1, the proposed system follows a fairly standard encoder-decoder architecture with an attention mechanism (Bahdanau et al., 2014). More concretely, we use a two-layer bidirectional RNN in the encoder, and another two-layer RNN in the decoder. All RNNs use GRU cells with 600 hidden units (Cho et al., 2014), and the dimensionality of the embeddings is set to 300. As for the attention mechanism, we use the global attention method proposed by Luong et al. (2015b) with the general alignment function. There are, however, three important aspects in which

our system differs from the standard NMT, and these are critical so the system can be trained in an unsupervised manner as described next in Section 3.2:

1. **Dual structure**. While NMT systems are typically built for a specific translation direction (e.g. either French → English or English → French), we exploit the dual nature of machine translation (He et al., 2016; Firat et al., 2016a) and handle both directions together (e.g. French ↔ English).

2. **Shared encoder**. Our system makes use of one and only one encoder that is shared by both languages involved, similarly to Ha et al. (2016), Lee et al. (2017) and Johnson et al. (2017). For instance, the exact same encoder would be used for both French and English. This universal encoder is aimed to produce a language independent representation of the input text, which each decoder should then transform into its corresponding language.

3. **Fixed embeddings in the encoder**. While most NMT systems randomly initialize their embeddings and update them during training, we use pre-trained cross-lingual embeddings in the encoder that are kept fixed during training. This way, the encoder is given language independent word-level representations, and it only needs to learn how to compose them to build representations of larger phrases. As discussed in Section 2.1, there are several unsupervised methods to train these cross-lingual embeddings from monolingual corpora, so this is perfectly feasible in our scenario. Note that, even if the embeddings are cross-lingual, we use separate vocabularies for each language. This way, the word *chair*, which exists both in French and English (meaning "flesh" in the former), would get a different vector in each language, although they would both be in a common space.

## 3.2 UNSUPERVISED TRAINING

As NMT systems are typically trained to predict the translations in a parallel corpus, such supervised training procedure is infeasible in our scenario, where we only have access to monolingual corpora. However, thanks to the architectural modifications proposed above, we are able to train the entire system in an unsupervised manner using the following two strategies:

1. **Denoising**. Thanks to the use of a shared encoder, and exploiting the dual structure of machine translation, the proposed system can be directly trained to reconstruct its own input. More concretely, the whole system can be optimized to take an input sentence in a given language, encode it using the shared encoder, and reconstruct the original sentence using the decoder of that language. Given that we use pre-trained cross-lingual embeddings in the shared encoder, this encoder should learn to compose the embeddings of both languages in a language-independent fashion, and each decoder should learn to decompose this representation into their corresponding language. At inference time, we simply replace the decoder with that of the target language, so it generates the translation of the input text from the language-independent representation given by the encoder.

    Nevertheless, this ideal behavior is severely compromised by the fact that the resulting training procedure is essentially a trivial copying task. As such, the optimal solution for this task would not need to capture any real knowledge of the languages involved, as there would be many degenerated solutions that blindly copy all the elements in the input sequence. If this were the case, the system would best make very literal word-by-word substitutions when used to translate from one language to another at inference time.

    In order to avoid such degenerated solutions and make the encoder truly learn the compositionality of its input words in a language independent manner, we propose to introduce random noise in the input sentences. The idea is to exploit the same underlying principle of denoising autoencoders (Vincent et al., 2010), where the system is trained to reconstruct the original version of a corrupted input sentence (Dai & Le, 2015; Hill et al., 2016). For that purpose, we alter the word order of the input sentence by making random swaps between contiguous words. More concretely, for a sequence of $N$ elements, we make $N/2$ random swaps of this kind. This way, the system needs to learn about the internal structure of the languages involved to be able to recover the correct word order. At the same time, by discouraging the system to rely too much on the word order of the input sequence, we can better account for the actual word order divergences across languages. This training procedure can be seen as an instance of contrastive estimation (Smith & Eisner, 2005), where the

neighborhood is defined by local swaps in our case, although other functions would also be possible.

2. **On-the-fly backtranslation**. In spite of the denoising strategy, the training procedure above is still a copying task with some synthetic alterations that, most importantly, involves a single language at each time, without considering our final goal of translating between two languages. In order to train our system in a true translation setting without violating the constraint of using nothing but monolingual corpora, we propose to adapt the backtranslation approach proposed by Sennrich et al. (2016a) to our scenario. More concretely, given an input sentence in one language, we use the system in inference mode with greedy decoding to translate it to the other language (i.e. apply the shared encoder and the decoder of the other language). This way, we obtain a pseudo-parallel sentence pair, and train the system to predict the original sentence from this synthetic translation.

Note that, contrary to standard backtranslation, which uses an independent model to backtranslate the entire corpus at one time, we take advantage of the dual structure of the proposed architecture to backtranslate each mini-batch on-the-fly using the model that is being trained itself. This way, as training progresses and the model improves, it will produce better synthetic sentence pairs through backtranslation, which will serve to further improve the model in the following iterations.

During training, we alternate these different training objectives from mini-batch to mini-batch. This way, given two languages L1 and L2, each iteration would perform one mini-batch of denoising for L1, another one for L2, one mini-batch of on-the-fly backtranslation from L1 to L2, and another one from L2 to L1. Moreover, by further assuming that we have access to a small parallel corpus, the system can also be trained in a semi-supervised fashion by combining these steps with directly predicting the translations in this parallel corpus just as in standard NMT.

## 4 EXPERIMENTAL SETTINGS

We make our experiments comparable with previous work by using the French-English and German-English **datasets** from the WMT 2014 shared task[2]. Following common practice, the systems are evaluated on newstest2014 using tokenized BLEU scores as computed by the `multi-bleu.perl` script[3]. As for the training data, we test the proposed system under three different settings:

- **Unsupervised**: This is the main scenario under consideration in our work, where the system has access to nothing but monolingual corpora. For that purpose, we used the News Crawl corpus with articles from 2007 to 2013.

- **Semi-supervised**: We assume that, in addition to monolingual corpora, we also have access to a small in-domain parallel corpus. This scenario has a great practical interest, as we might often have some parallel data from which we could potentially benefit, but it is insufficient to train a full traditional NMT system. For that purpose, we used the same monolingual data from the unsupervised settings together with either 10,000 or 100,000 random sentence pairs from the News Commentary parallel corpus.

- **Supervised**: This is the traditional scenario in NMT where we have access to a large parallel corpus. While not the focus of our work, this setting should provide an approximate upper-bound for the proposed system. For that purpose, we used the combination of all parallel corpora provided at WMT 2014, which comprise Europarl, Common Crawl and News Commentary for both language pairs plus the UN and the Gigaword corpus for French-English. For direct comparison with the semi-supervised scenario, we also ran separate experiments using the same subsets of News Commentary alone.

Note that, to be faithful to our target scenario, we did not make use of any parallel data in these language pairs for development or tuning purposes. Instead, we used Spanish-English WMT data for our preliminary experiments, where we also decided all the hyperparameters without any rigorous exploration.

---

[2] http://www.statmt.org/wmt14/translation-task.html
[3] https://github.com/moses-smt/mosesdecoder/blob/master/scripts/generic/multi-bleu.perl

As for the **corpus preprocessing**, we perform tokenization and truecasing using standard Moses tools.[4] We then apply byte pair encoding (BPE) as proposed by Sennrich et al. (2016b) using the implementation provided by the authors[5]. Learning was done on the monolingual corpus of each language independently, using 50,000 operations. While BPE is known to be an effective way to overcome the rare word problem in standard NMT, it is less clear how it would perform in our more challenging unsupervised scenario, as it might be difficult to learn the translation relations between subword units. For that reason, we also run experiments at the word level in this unsupervised scenario, limiting the vocabulary to the most frequent 50,000 tokens and replacing the rest with a special token <UNK>. We accelerate training by discarding all sentences with more than 50 elements (either BPE units or actual tokens).

Given that the proposed system uses pre-trained **cross-lingual embeddings** in the encoder as described in Section 3.1, we use the monolingual corpora described above to independently train the embeddings for each language using word2vec (Mikolov et al., 2013b). More concretely, we use the skip-gram model with ten negative samples, a context window of ten words, 300 dimensions, a sub-sampling of $10^{-5}$, and ten training iterations. We then use the public implementation[6] of the method proposed by Artetxe et al. (2017) to map these embeddings to a shared space, using the recommended configuration with numeral-based initialization. In addition to being a component of the proposed system, the resulting embeddings are also used to build a simple **baseline system** that translates a sentence word-by-word, replacing each word by their nearest neighbor in the other language and leaving out-of-vocabularies unchanged.

The **training** of the proposed system itself is done using the procedure described in Section 3.2 with the cross-entropy loss function and a batch size of 50 sentences. For the unsupervised systems, we try using denoising alone as well as the combination of both denoising and backtranslation, in order to better analyze the contribution of the latter. We use Adam as our optimizer with a learning rate of $\alpha = 0.0002$ (Kingma & Ba, 2015). During training, we use dropout regularization with a drop probability $p = 0.3$. Given that we restrict ourselves not to use any parallel data for development purposes, we perform a fixed number of iterations (300,000) to train each variant. Using our PyTorch implementation, training each system took about 4-5 days on a single Titan X GPU for the full unsupervised variant. Although we observed that the system had not fully converged after this number of iterations in our preliminary experiments, we decide to stop training at this point in order to accelerate experimentation due to hardware constraints.

As described in Section 3.2, we use greedy **decoding** at training time for backtranslation, but actual inference at test time was done using beam-search with a beam size of 12 following common practice (Sutskever et al., 2014; Sennrich et al., 2016a;b; He et al., 2016). We do not use any length or coverage penalty, which might further improve the reported results.

## 5 RESULTS AND DISCUSSION

We discuss the quantitative results in Section 5.1, and present a qualitative analysis in Section 5.2.

### 5.1 QUANTITATIVE ANALYSIS

The BLEU scores obtained by all the tested variants are reported in Table 1.

As it can be seen, the proposed **unsupervised system** obtains very strong results considering that it was trained on nothing but monolingual corpora, reaching 14-15 BLEU points in French-English and 6-10 BLEU points in German-English depending on the variant and direction (rows 3 and 4). This is much stronger than the baseline system of word-by-word substitution (row 1), with improvements of at least 40% in all cases, and up to 140% in some (e.g. from 6.25 to 15.13 BLEU points in English $\rightarrow$ French). This shows that the proposed system is able to go beyond very literal translations, effectively learning to use context information and account for the internal structure of the languages.

The results also show that **backtranslation** is essential for the proposed system to work properly. In fact, the denoising technique alone is below the baseline (row 1 vs 2), while big improvements are

---

[4]https://github.com/moses-smt/mosesdecoder
[5]https://github.com/rsennrich/subword-nmt
[6]https://github.com/artetxem/vecmap

Table 1: BLEU scores in newstest2014. Unsupervised systems are trained in the News Crawl monolingual corpus, semi-supervised systems are trained in the News Crawl monolingual corpus and a subset of the News Commentary parallel corpus, and supervised systems (provided for comparison) are trained in either these same subsets or the full parallel corpus, all from WMT 2014. For GNMT, we report the best single model scores from Wu et al. (2016).

|  |  | FR-EN | EN-FR | DE-EN | EN-DE |
|---|---|---|---|---|---|
| **Unsupervised** | 1. Baseline (emb. nearest neighbor) | 9.98 | 6.25 | 7.07 | 4.39 |
|  | 2. Proposed (denoising) | 7.28 | 5.33 | 3.64 | 2.40 |
|  | 3. Proposed (+ backtranslation) | 15.56 | 15.13 | 10.21 | 6.55 |
|  | 4. Proposed (+ BPE) | 15.56 | 14.36 | 10.16 | 6.89 |
| **Semi-supervised** | 5. Proposed (full) + 10k parallel | 18.57 | 17.34 | 11.47 | 7.86 |
|  | 6. Proposed (full) + 100k parallel | 21.81 | 21.74 | 15.24 | 10.95 |
| **Supervised** | 7. Comparable NMT (10k parallel) | 1.88 | 1.66 | 1.33 | 0.82 |
|  | 8. Comparable NMT (100k parallel) | 10.40 | 9.19 | 8.11 | 5.29 |
|  | 9. Comparable NMT (full parallel) | 20.48 | 19.89 | 15.04 | 11.05 |
|  | 10. GNMT (Wu et al., 2016) | - | 38.95 | - | 24.61 |

seen when introducing backtranslation (row 2 vs 3). Test perplexities also confirm this: for instance, the proposed system with denoising alone obtains a per-word perplexity of 634.79 for French → English, whereas the one with backtranslation achieves a much lower perplexity of 44.74. We emphasize, however, that the proposed training procedure would not work using backtranslation alone without denoising, as the initial translations would be meaningless sentences produced by a random NMT model, encouraging the system to completely ignore the input sentence and simply learn a language model of the target language. We thus conclude that both denoising and backtranslation play an essential role during training: denoising forces the system to capture broad word-level equivalences, while backtranslation encourages it to learn more subtle relations in an increasingly natural setting.

As for the role of **subword** translation, we observe that **BPE** is slightly beneficial when German is the target language, detrimental when French is the target language, and practically equivalent when English is the target language (row 3 vs 4). This might be a bit surprising considering that the word-level system does not handle out-of-vocabularies in any way, so it always fails to translate rare words. Having a closer look, however, we observe that, while BPE manages to correctly translate some rare words, it also introduces some new errors. In particular, it sometimes happens that a subword unit from a rare word gets prefixed to a properly translated word, yielding to translations like *SevAgency* (split as *S- ev- Agency*). Moreover, we observe that BPE is of little help when translating infrequent named entities. For instance, we observed that our system translated *Tymoshenko* as *Ebferchenko* (split as *Eb- fer- chenko*). While standard NMT would easily learn to copy this kind of named entities using BPE, such relations are much more challenging to model under our unsupervised learning procedure. This way, we believe that a better handling of rare words and, in particular, named entities and numerals, could further improve the results in the future.

In addition to that, the results of the **semi-supervised system** (rows 5 and 6) show that the proposed model can greatly benefit from a small parallel corpus. Note that these semi-supervised systems differ from the full unsupervised system (row 4) in the use of either 10,000 or 100,000 parallel sentences from News Crawl, so that their training alternates between denoising, backtranslation and, additionally, maximizing the translation probability of these parallel sentences as described in Section 3.2. As it can be seen, 10,000 parallel sentences alone bring an improvement of 1-3 BLEU points, while 100,000 sentences bring an improvement of 4-7 points. These results are much better than those of a comparable NMT system trained in the same parallel data (rows 7 and 8), showing the potential interest of our approach beyond the strictly unsupervised scenario. In fact, the semi-supervised system trained in 100,000 parallel sentences (row 6) even surpasses the comparable NMT system trained in the full parallel corpus (row 9) in all cases but one, presumably because the domain of both the monolingual and the parallel corpora that it uses matches that of the test set.

Table 2: Sample French→English translations from newstest2014 by the full proposed system with BPE. See text for comments.

| Source | Reference | Proposed system (full) |
|---|---|---|
| Une fusillade a eu lieu à l'aéroport international de Los Angeles. | There was a shooting in Los Angeles International Airport. | A shooting occurred at Los Angeles International Airport. |
| Cette controverse croissante autour de l'agence a provoqué beaucoup de spéculations selon lesquelles l'incident de ce soir était le résultat d'une cyber-opération ciblée. | Such growing controversy surrounding the agency prompted early speculation that tonight's incident was the result of a targeted cyber operation. | This growing scandal around the agency has caused much speculation about how this incident was the outcome of a targeted cyber operation. |
| Le nombre total de morts en octobre est le plus élevé depuis avril 2008, quand 1 073 personnes avaient été tuées. | The total number of deaths in October is the highest since April 2008, when 1,073 people were killed. | The total number of deaths in May is the highest since April 2008, when 1 064 people had been killed. |
| À l'exception de l'opéra, la province reste le parent pauvre de la culture en France. | With the exception of opera, the provinces remain the poor relative of culture in France. | At an exception, opera remains of the state remains the poorest parent culture. |

As for the **supervised system**, it is remarkable that the comparable NMT model (rows 7-9), which uses the proposed architecture but trains it to predict the translations in the corresponding parallel corpus, obtains poor results compared to the state of the art in NMT (e.g. GNMT in row 10). Note that the comparable NMT system is equivalent to the semi-supervised system (rows 5 and 6), except that it does not use any monolingual corpora nor, consequently, denoising and backtranslation. As such, the comparable NMT differs from standard NMT in the use of a shared encoder with fixed embeddings (Section 3.1) and input corruption (Section 3.2).

The relatively poor results of the comparable NMT model suggest that these additional constraints in our system, which were introduced to enable unsupervised learning, may also be a factor limiting its potential performance, so we believe that the system **could be further improved in the future** by progressively relaxing these constraints during training. For instance, using fixed cross-lingual embeddings in the encoder is necessary in the early stages of training, as it forces the encoder to use a common word representation for both languages, but it might also limit what it can ultimately learn in the process. For that reason, one could start to progressively update the weights of the encoder embeddings as training progresses. Similarly, one could also decouple the shared encoder into two independent encoders at some point during training, or progressively reduce the noise level. At the same time, note that we did not perform any rigorous hyperparameter exploration, and favored efficiency over performance in the experimental design due to hardware constraints. As such, we think that there is a considerable margin to improve these results by using larger models, longer training times, and incorporating several well-known NMT techniques (e.g. ensembling and length/coverage penalty).

## 5.2 QUALITATIVE ANALYSIS

In order to better understand the behavior of the proposed system, we manually analyzed some translations for French → English, and present some illustrative examples in Table 2.

Our analysis shows that the proposed system is able to produce high-quality translations, adequately modeling non-trivial translation relations. For instance, in the first example it translates the expression *a eu lieu* (literally "has had place") as *occurred*, going beyond a literal word-by-word substitution. At the same time, it correctly translates *l'aéroport international de Los Angeles* as *Los Angeles International Airport*, properly modeling structural differences between the languages. As shown by the second example, the system is also capable of producing high-quality translations for considerably longer and more complex sentences.

Nevertheless, our analysis also points that the proposed system has limitations and, perhaps not surprisingly, its translation quality often lags behind that of a standard supervised NMT system. In particular, we observe that the proposed model has difficulties to preserve some concrete details from source sentences. For instance, in the third example *April* and *2008* are properly translated, but *octobre* (”October”) is mistranslated as *May* and *1 073* as *1 064*. While these clearly point to some adequacy issues, they are also understandable given the unsupervised nature of the system, and it is remarkable that the system managed to at least replace a month by another month and a number by another close number. We believe that incorporating character level information might help to mitigate some of these issues, as it could for instance favor *October* as the translation of *octobre* instead of the selected *May*.

Finally, there are also some cases where there are both fluency and adequacy problems that severely hinders understanding the original message from the proposed translation. For instance, in the last example our system preserves most keywords in the original sentence, but it would be difficult to correctly guess its meaning just by looking at its translation. In concordance with our quantitative analysis, this suggests that there is still room for improvement, opening new research avenues for the future.

## 6    CONCLUSIONS AND FUTURE WORK

In this work, we propose a novel method to train an NMT system in a completely unsupervised manner. We build upon existing work on unsupervised cross-lingual embeddings (Artetxe et al., 2017; Zhang et al., 2017), and incorporate them in a modified attentional encoder-decoder model. By using a shared encoder with these fixed cross-lingual embeddings, we are able to train the system from monolingual corpora alone, combining denoising and backtranslation.

The experiments show the effectiveness of our proposal, obtaining significant improvements in the BLEU score over a baseline system that performs word-by-word substitution in the standard WMT 2014 French-English and German-English benchmarks. Our manual analysis confirms the quality of the proposed system, showing that it is able to model complex cross-lingual relations and produce high-quality translations. Moreover, we show that combining our method with a small parallel corpus can bring further improvements, showing its potential interest beyond the strictly unsupervised scenario.

Our work opens exciting opportunities for future research, as our analysis reveals that, in spite of the solid results, there is still a considerable room for improvement. In particular, we observe that the performance of a comparable supervised NMT system is considerably below the state of the art, which suggests that the architectural modifications introduced by our proposal (Section 3.1) are also limiting its potential performance. For that reason, we would like to explore progressively relaxing these constraints during training as discussed in Section 5.1. Additionally, we would like to incorporate character level information into the model, which we believe that could be very helpful to address some of the adequacy issues observed in our manual analysis (Section 5.2). Finally, we would like to explore other neighborhood functions for denoising, and analyze their effect in relation to the typological divergences of different language pairs.

### ACKNOWLEDGMENTS

This research was partially supported by a Google Faculty Award, the Spanish MINECO (TUNER TIN2015-65308-C5-1-R, MUSTER PCIN-2015-226 and TADEEP TIN2015-70214-P, cofunded by EU FEDER), the Basque Government (MODELA KK-2016/00082), the UPV/EHU (excellence research group), and the NVIDIA GPU grant program. Mikel Artetxe enjoys a doctoral grant from the Spanish MECD. Kyunghyun Cho thanks support by eBay, TenCent, Facebook, Google, NVIDIA and CIFAR, and was partly supported by Samsung Advanced Institute of Technology (Next Generation Deep Learning: from pattern recognition to AI).

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
