# OpenReview forum: "Unsupervised Neural Machine Translation"
_ICLR.cc/2018/Conference — Accept (Poster)_

### Official Review · AnonReviewer3 · 2017-11-27
**Very few original ideas**

**Rating:** 6
**Confidence:** 4

**Review:**

The authors present a model for unsupervised NMT which requires no parallel corpora between the two languages of interest. While the results are interesting I find very few original ideas in this paper. Please find my comments/questions/suggestions below:

1) The authors mention that there are 3 important aspects in which their model differs from a standard NMT architecture. All the 3 differences have been adapted from existing works. The authors clearly acknowledge and cite the sources. Even sharing the encoder using cross lingual embeddings has been explored in the context of multilingual NER (please see https://arxiv.org/abs/1607.00198). Because of this I find the paper to be a bit lacking on the novelty quotient. Even backtranslation has been used successfully in the past (as acknowledged by the authors). Unsupervised MT in itself is not a new idea (again clearly acknowledged by the authors).

2) I am not very convinced about the idea of denoising. Specifically, I am not sure if it will work for arbitrary language pairs. In fact, I think there is a contradiction even in the way the authors write this. On one hand, they want to "learn the internal structure of the languages involved" and on the other hand they deliberately corrupt this structure by adding noise. This seems very counter-intuitive and in fact the results in Table 1 suggest that it leads to a drop in performance. I am not very sure that the analogy with autoencoders holds in this case.

3) Following up on the above question, the authors mention that "We emphasize, however, that it is not possible to use backtranslation alone without denoising". Again, if denoising itself leads to a drop in the performance as compared to the nearest neighbor baseline then why use backtranslation in conjunction with denoising and not in conjunction with the baseline itself.

4) This point is more of a clarification and perhaps due to my lack of understanding. Backtranslation to generate a pseudo corpus makes sense only after the model has achieved a certain (good) performance. Can you please provide details of how long did you train the model (with denoising?) before producing the backtranslations ?

5) The authors mention that 100K parallel sentences may be insufficient for training a NMT system. However, this size may be decent enough for  a PBSMT system. It would be interesting to see the performance of a PBSMT system trained on 100K parallel sentences.

6) How did you arrive at the beam size of 12 ? Was this a hyperparameter? Just curious.

7) The comparable NMT set up is not very clear. Can you please explain it in detail ? In the same paragraph, what exactly do you mean by "the supervised system in this paper is relatively small?"

---

> ### Author Response · Authors · 2017-12-24
> **Re: Very few original ideas**
>
> Thanks for the insightful comments. Please find the answers to the specific points below, which were also addressed in the revised version of the paper:
>
> 1) We are aware that the basic building blocks of our work come from previous work and, as you note, we try to properly acknowledge that in the paper. However, we do not see this as a weakness, but rather an inherent characteristic of science as a collaborative effort. Our contribution lies in combining these basic building blocks in a novel way to build the first fully unsupervised NMT system. We believe that this is an important contribution on its own: NMT is a highly relevant field where the predominant approach has been supervised and, for the first time, we show that an unsupervised approach is also viable. As such, we think that our work explores a highly original idea and opens a new and exciting research direction.
>
> 2) This is a very interesting observation, but we think that, paradoxically, corrupting the structure of the language is necessary for the system to learn such structure. Note that, without denoising, this training step would be reduced to a trivial copying task that admits degenerated solutions. The intuition is that one can easily copy a sentence in any language even if they know nothing about that language. In contrast, adding noise to the input makes the task of reconstructing the input non-trivial, and forces the system to learn about the structure of that language to be able to solve it. The intuition in this case is that, if we are given a scrambled sentence in some language, it is not possible for us to recover the original sentence unless we have some knowledge of the language in question. In other words, the idea of denoising is to corrupt the structure of the input, so the system needs to learn the correct structure in order to recover the original uncorrupted input (this is possible because the system does see the correct structure in the output during training). Note that this has also been found to help extract good representations from natural language sentences by other authors (Hill et al., 2016). Regarding arbitrary language pairs, we think that this idea is particularly relevant for distant pairs: corrupting the word order of the input makes the shared encoder rely less in this word order, which is necessary for distant language pairs with more divergences in this regard.
>
> 3/4) There seems to be some confusion here on how our training procedure works in relation to backtranslation. Note that each training iteration performs one mini-batch of denoising and one mini-batch of backtranslation in each direction, and the bactranslation step at iteration i uses the model from iteration (i-1). This way, denoising and backtranslation keep constantly improving the model, and backtranslation itself uses the most recent model at each time. This is in contrast with traditional backtranslation, where a fixed model is used to backtranslate the entire corpus at one time. In relation to point 3, while denoising alone is certainly weaker than the nearest neighbor baseline, the combination of denoising and backtransation eventually leads to a stronger model, which backtranslation itself takes advantage of in the following iterations as just described.
>
> 5) The purpose of this experiment was to show that the proposed system can also benefit from small parallel corpora, making it suitable not only for the unsupervised scenario, but also for the semi-supervised scenario. As such, our point was not to improve the state of the art under these conditions, but rather to show that our work has potential interest beyond the strictly unsupervised scenario.
>
> 6) A beam size of 12 is very common in previous work (Sutskever et al., 2014; Sennrich et al., 2016a;b; He et al., 2016), so we also adopted it for our experiments without any further exploration.
>
> 7) The comparable NMT system uses the exact same architecture and hyperparameters as the unsupervised system (number of layers, hidden units, attention model etc.). Furthermore, it incorporates the non-standard variations in Section 3.1 (dual structure using a shared encoder with fixed embeddings). The only difference is that, instead of training it on monolingual corpora using denoising and backtranslation, it is trained on parallel corpora just as standard NMT.
>
> When we say that "the supervised system in this paper is relatively small", we mean that the size of the model (number of layers, training time etc.) is small compared to the state of the art, which explains in part why its results are also weaker. However, note that this also applies to the unsupervised system, which uses the exact same settings. We therefore believe that there is a considerable margin to improve our results by using a larger model.

---

### Official Review · AnonReviewer1 · 2017-11-27
**Interesting but preliminary proof-of-concept**

**Rating:** 5
**Confidence:** 4

**Review:**

unsupervised neural machine translation

This is an interesting paper on unsupervised MT. It trains a standard architecture using:

1) word embeddings in a shared embedding space, learned using a recent approach that works with only tens of bilingual word papers.

2) A encoder-decoder trained using only monolingual data (should cite http://www.statmt.org/wmt17/pdf/WMT15.pdf). Training uses a “denoising” method which is not new: it uses the same idea as contrastive estimation (http://www.aclweb.org/anthology/P05-1044, a well-known method which should be cited).

3) Backtranslation.

All though none of these ideas are new, they haven’t been combined in this way before, and that what’s novel here. The paper is essentially a neat application of (1), and is an empirical/ systems paper. It’s essentially a proof-of-concept that it is that it’s possible to get anything at all using no parallel data. That’s surprising and interesting, but I learned very little else from it. The paper reads as preliminary and rushed, and I had difficulty answering some basic questions:

* In Table (1), I’m slightly puzzled by why 5 is better than 6, and this may be because I’m confused about what 6 represents. It would be natural to compare 5 with a system trained on 100K parallel text, since the systems would then (effectively) differ only in that 5 also exploits additional monolingual data. But the text suggests that 6 is trained on much more than 100K parallel sentences; that is, it differs in at least two conditions (amount of parallel text and use of monolingual text). Since this paper’s primary contribution is empirical, this comparison should be done in a carefully controlled way, differing each of these elements in turn.

* I’m very confused by the comment on p. 8 that “the modifications introduced by our proposal are also limiting” to the “comparable supervised NMT system”. According to the paper, the architecture of the system is unchanged, so why would this be the case? This comment makes it seem like something else has been changed in the baseline, which in turn makes it somewhat hard to accept the results here.

Comment:
* The qualitative analysis is not really an analysis: it’s just a few cherry-picked examples and some vague observations. While it is useful to see that the system does indeed generate nontrivial content in these cases, this doesn’t give us further insight into what the system does well or poorly outside these examples. The BLEU scores suggest that it also produces many low-quality translations. What is different about these particular examples? (Aside: since the cross-lingual embedding method is trained on numerals, should we be concerned that the system fails at translating numerals?)

Questions:
* Contrastive estimation considers other neighborhood functions (“random noise” in the parlance of this paper), and it’s natural to wonder what would happen if this paper also used these or other neighborhood functions. More importantly, I suspect the the neighborhood functions are important: when translating between Indo-European languages as in these experiments, local swaps are reasonable; but in translating between two different language families (as would often be the case in the motivating low-resource scenario that the paper does not actually test), it seems likely that other neighborhood functions would be important, since structural differences would be much larger.

Presentational comments (these don’t affect my evaluation, they’re mostly observations but they contribute to a general feeling that the paper is rushed and preliminary):

* BPE does not “learn”, it’s entirely deterministic.

* This paper is at best tangentially related to decipherment. Decipherment operates under two quite different assumptions: there is no training data for the source language ciphertext, only the ciphertext itself (which is often very small); and the replacement function is deterministic rather than probabilistic (and often monotonic). The Dou and Knight papers are interesting, but they’re an adaptation of ideas rather than decipherment per se. Since none of those ideas are used here this feels like hand-waving.

* Future work is vague: “we would like to detect and mitigate the specific causes…” “we also think that a better handling of rare words…” That’s great, but how will you do these things? Do you have specific reasons to think this, or ideas on how to approach them? Otherwise this is just hand-waving.

---

> ### Author Response · Authors · 2017-12-24
> **Re: Interesting but preliminary proof-of-concept (part 1/2)**
>
> Thanks for the insightful review. We have tried to make the paper more clear in the revised version taking these comments into account. Please find the answers to each specific point below:
>
> General:
>
> - To clarify what the semi-supervised and supervised systems represents in Table 1: (5) is the same as (4), but in addition to training on monolingual corpora using denoising and backtransaltion, it is also trained in a subset of 100K parallel sentences using standard supervised cross-entropy loss (it alternates one mini-batch of denoising, one mini-batch of backtranslation and one mini-batch of  this supervised training). (6) is the same as (5) except for two differences: 1) it uses the full parallel corpus instead of the subset of 100K parallel sentences, and 2) it does not use any monolingual corpora nor denoising or backtranslation. We think that the main reason why (5) is better than (6) is related to the domain: the parallel corpus of (6) is general, whereas the subset of 100K parallel sentences and the monolingual corpus used for (5) are in the news domain just as the test set. While these facts were already mentioned in the paper, the new version includes a more detailed discussion. At the same time, we agree that the fact that the comparable system differs from the semi-supervised system in two aspects makes the comparison more difficult, and we are currently working to extend our experiments accordingly.
>
> - Regarding the comment that “the modifications introduced by our proposal are also limiting” to the “comparable supervised NMT system”, note that, as discussed in the previous point, the comparable NMT system uses the exact same architecture and hyperparameters as the unsupervised system (number of layers, hidden units, attention model etc.) and, as such, it also incorporates the non-standard variations in Section 3.1 (dual structure using a shared encoder with fixed embeddings). These are what we were referring to as “the modification introduced by our proposal”, but the only difference between the unsupervised and the supervised systems is that, instead of training in monolingual corpora using denoising and backtranslation, we train in parallel corpora just as in standard NMT. We have tried to make this more clear in the revised version of the paper.
>
> Comment:
>
> - We agree that the qualitative analysis in the current version is limited. It was done mainly to check and illustrate that the proposed unsupervised NMT system generates sensible translations despite the lack of parallel corpora. We believe that a more detailed investigation and analysis into the properties and characteristics of translation generated by unsupervised NMT must be conducted in the future.
>
> - Note that we only use shared numerals to initialize the iterative embedding mapping method, so it is understandable that the system fails to translate numerals after the training of both the embedding mapping and the unsupervised NMT system itself. While it would certainly be possible to perform an ad-hoc processing to translate numerals under the assumption that they are shared by different languages, our point was to show that the system has some logical adequacy issues for very similar concepts (e.g. different numerals or month names).
>
> Questions:
>
> - Thanks for pointing out the connection with contrastive estimation, which is now properly discussed in the revised version of the paper. As for the role of neighborhood functions, we agree that there are many possible choices beyond local swaps, and the optimal choice could greatly depend on the typological divergences between the languages involved. In this regard, we think that this is a very interesting direction to explore in the future, and we have tried to better discuss this matter in the revised version of the paper.
>
> Having said that, please note that we have considered two language pairs (English-French and English-German) in our experiments. Despite being indo-european, there are important properties that distinguish these language pairs, such as the verb-final construction and the prevalence of compounding in German in contrast to French. In fact, English-German has often been studied in machine translation as a particularly challenging language pair. For that reason, we believe that the experiments on these two distinct language pairs support the effectiveness of the proposed approach, despite the potential for future investigation on the effect of contrastive functions on the choice of language pairs.

---

> > ### Author Response · Authors · 2017-12-24
> > **Re: Interesting but preliminary proof-of-concept (part 2/2)**
> >
> > Presentational comments:
> >
> > - We are aware that BPE is completely deterministic. However, it does require to extract some merge operations that are later applied. The original paper of Sennrich et al. (2016) refers to this process as "learning", so we decided to follow their wording.
> >
> > - The Dou and Knight papers also attempt to build machine translation systems using monolingual corpora, so we briefly discuss and acknowledge their work accordingly even if our approach is completely different. Regarding the choice of the term "decipherment" to refer to that work, we understand that this might not exactly adjust to the common acceptation of the term, but it seems to be the one that the authors themselves use (e.g. "Unifying bayesian inference and vector space models for improved decipherment"). We have rewritten it as "statistical decipherment for machine translation", which we hope that is more descriptive.
> >
> > - We have rewritten the future work in the revised version of the paper trying to be more specific.

---

> > ### Comment · AnonReviewer1 · 2018-01-11
> > **Response re: questions about experiments.**
> >
> > Thanks for the clarifications. They raise some new questions for me.
> >
> > The revision states (at the bottom of p.7) that the constraints (i.e. fixed cross-lingual embeddings) are necessary for learning. This seems like something that could only be discovered empirically, but the evidence is not in the paper. Do you have other experiments that show this? As discussed elsewhere in these reviews, the contribution of this paper is empirical, and IMO it would be *much* stronger if it included systematic experiments that empirically probe the importance of its architectural choices.
> >
> > Of course, if these changes cripple the comparable NMT system then that baseline is only part of the story. What most readers will want to know is: how does this compare to a standard supervised NMT system (and SMT system, as pointed out by the other reviewers), since that’s the real choice that a practitioner would be faced with. Do you have experiments on this?
> >
> > That these answers aren't in the paper reinforce my feeling that the paper is preliminary. It would be much more convincing if it presented empirical evidence and toned down the rhetoric. Show, don't tell.

---

> > > ### Author Response · Authors · 2018-01-12
> > > **Re: Questions about experiments**
> > >
> > > To put things into perspective, we would like to remark that, as reflected by the title itself, the focus of our work is on unsupervised NMT. To the best of our knowledge, this is the first working approach to train an NMT system with nothing but monolingual corpora (concurrently with another submission at https://openreview.net/forum?id=rkYTTf-AZ), and this is in fact where our main contribution lies in.
> > >
> > > For that reason, we disagree that "what most readers will want to know is: how does this compare to a standard supervised NMT system [...] since that's the real choice that a practitioner would be faced with". Needless to say, "a standard supervised NMT system" cannot work in an unsupervised scenario. As said before, this is the main focus of our work, and there is no such choice to be made in it, as standard machine translation is not an option when we only have access to monolingual corpora.
> > >
> > > Besides that, we think that our experiments already give a reasonably complete picture of how the proposed system compares to standard supervised approaches. We report the results of a comparable NMT system for different corpora sizes, as well as those of GNMT, which can be taken as a representative state-of-the-art machine translation system. Moreover, we run our experiments in a widely used dataset, making our results easily comparable to other work.
> > >
> > > Finally, we acknowledge that our work is not necessarily the best possible approach to unsupervised NMT. It is, however, the first working approach to unsupervised NMT, and this is where our main contribution lies in. As such, it is to be expected that significant advances will be made in the future, but we do not think that this makes our work preliminary. We think that our approach is well motivated, the experiments convincingly show its solid performance and, overall, our work sets a strong foundation for a new and exciting research direction in NMT.

---

> > ### Comment · AnonReviewer1 · 2018-01-11
> > **Re: English and German**
> >
> > It's true that certain aspects of English-German translation are difficult, and indeed, phrase-based MT is notoriously bad at the verb-final construction, often failing to translate the verb altogether. But the paper doesn't analyse the performance of these systems on this specific phenomenon. If it did, it would be a much stronger analysis than what's currently in the paper. You can get reasonable BLEU scores in English-German with just a word-for-word gloss, so, in the absence of any analysis, I would hypothesize that the system does something like this.

---

> > > ### Author Response · Authors · 2018-01-12
> > > **Re: English and German**
> > >
> > > While it is true that we do not analyze any specific linguistic phenomenon in depth, note that our experiments already show that the system is not working like a "word-for-word gloss" as speculated in the comment: the baseline system is precisely a word-for-word gloss, and the proposed method beats it with a considerable margin. Note that this baseline system is based on the exact same cross-lingual embeddings as the proposed system, meaning that our approach must have learned non-trivial translation relations that go beyond a word-for-word substitution to surpass it.
> > >
> > > In addition to that, the examples in Table 2 also show that the proposed method is able to properly handle word reordering and multiword expressions. As discussed in the paper, there are also instances where the system does a poor job in these aspects, but these examples confirm that the proposed approach is able to go beyond a word-for-word gloss.

---

### Official Review · AnonReviewer2 · 2017-11-27
**A straightforward first step toward unsupervised NMT**

**Rating:** 7
**Confidence:** 5

**Review:**

This paper describes a first working approach for fully unsupervised neural machine translation. The core ideas being this method are: (1) train in both directions (French to English and English to French) in tandem; (2) lock the embedding table to bilingual embeddings induced from monolingual data; (3) share the encoder between two languages; and (3)  alternate between denoising auto-encoder steps and back-translation steps. The key to making this work seems to be using a denoising auto-encoder where noise is introduced by permuting the source sentence, which prevents the encoder from learning a simple copy operation. The paper shows real progress over a simple word-to-word baseline for WMT 2014 English-French and English-German. Preliminary results in a semi-supervised setting are also provided.

This is solid work, presenting a reasonable first working system for unsupervised NMT, which had never been done before now. That alone is notable, and overall, I like the paper. The work shares some similarities with He et al.’s NIPS 2016 paper on “Dual learning for MT,” but has more than enough new content to address the issues that arise with the fully unsupervised scenario. The work is not perfect, though. I feel that the paper’s abstract over-claims to some extent. Also, the experimental section shows clearly that in getting the model to work at all, they have created a model with a very real ceiling on performance. However, to go from not working to working a little is a big, important first step. Also, I found the paper’s notation and prose to be admirably clear; the paper was very easy to follow.

Regarding over-claiming, this is mostly an issue of stylistic preference, but this paper’s use of the term “breakthrough” in both the abstract and the conclusion grates a little. This is a solid first attempt at a new task, and it lays a strong foundation for others to build upon, but there is lots of room for improvement. I don’t think it warrants being called a breakthrough - lots of papers introduce new tasks and produce baseline solutions. I would generally advise to let the readers draw their own conclusions.

Regarding the ceiling, the authors are very up-front about this in Table 1, but it bears repeating here: a fully supervised model constrained in the same way as this unsupervised model does not perform very well at all. In fact, it consistently fails to surpass the semi-supervised baseline (which I think deserved some further discussion in the paper). The poor performance of the fully supervised model demonstrates that there is a very real ceiling to this approach, and the paper would be stronger if the authors were able to show to what degree relaxing these constraints harms the unsupervised system and helps the supervised one.

The semi-supervised experiment in Sections 2.3 and 4 is a little dangerous. With BLEU scores failing to top 22 for English-French, there is a good chance that a simple phrase-based baseline on the same 100k sentence pairs with a large target language model will outperform this technique. Any low-resource scenario should include a Moses baseline for calibration, as NMT is notoriously weak with small amounts of parallel data.

Finally, I think the phrasing in Section 5.1 needs to be softened, where it states, “... it is not possible to use backtranslation alone without denoising, as the initial translations would be meaningless sentences produced by a random NMT model, ...” This statement implies that the system producing the sentences for back-translation must be a neural MT system, which is not the case. For example, a related paper co-submitted to ICLR, called “Unsupervised machine translation using monolingual corpora only,” shows that one can prime back-translation with a simple word-to-word system similar to the word-to-word baseline in this paper’s Table 1.

---

> ### Author Response · Authors · 2017-12-24
> **Re: A straightforward first step toward unsupervised NMT**
>
> Thanks for the insightful feedback. Please find our answers below:
>
> - Regarding over-claiming, it was not our intention to exaggerate our contribution, and we in fact share your view on this: we think that our work is a strong foundation for a new and exciting research direction in NMT, but we agree that it is only a first step and there is still a long way to go. We understand that “breakthrough” might not be the most appropriate term for this, and we have removed it from the revised version of the paper.
>
> - We find the discussion on the ceiling very interesting and relevant. We agree on the following key observations: 1) the comparable supervised system can be seen as a ceiling for the unsupervised system, and 2) the comparable supervised system gets relatively poor results. As such, one might conclude that our approach has a hard limit in this ceiling, as any eventual improvement in the proposed training method could at best close the gap with it. However, this also assumes that the ceiling itself is fixed and cannot be improved, which we do not find to be the case. In fact, we think that a very interesting research direction is to identify and address the factors that limited the performance of the comparable supervised system, which should also translate in an improvement for the unsupervised system. We have the following ideas in this regard, which we have better described in the revised version of the paper:
>
> 1) We did not perform any rigorous hyperparameter exploration, and we favored efficiency over performance in our experimental design. As such, we think that there is a considerable margin to improve our results with some engineering effort, such as using larger models, longer training times, ensembling techniques and better decoding strategies (length/coverage penalty).
>
> 2) While the constraints that we introduce to make our unsupervised system trainable might also limit its performance, one could design a multi-phase training procedure where these constraints are progressively relaxed. For instance, a key aspect of our design is to use fixed cross-lingual embeddings in the encoder. This is necessary in the early stages of training, as it forces the encoder to use a common word representation for both languages, but it might also limit what it can ultimately learn in the process. For that reason, one could start to progressively update the weights of the encoder embeddings as training progresses. Similarly, one could also decouple the shared encoder into two independent encoders at some point during training, or progressively reduce the noise level.
>
> - Regarding the semi-supervised experiments, note that our point here was not to improve the state of the art under these conditions, but rather to prove that the proposed system can also exploit a (relatively) small parallel corpus, showing its potential interest beyond the strictly unsupervised scenario.
>
> - Our statement that “it is not possible to use backtranslation alone without denoising” was referring to our training procedure where backtranslation uses the model from the previous iteration. It is true that it does not apply to the general case, as backtranslation could also be used in conjunction with other translation methods (e.g. embedding nearest neighbor), and we have consequently softened the statement in the revised version as suggested.

---

### Public Comment · ~Anthony_Chen1 · 2017-12-16
**Reproducibility of Denoising and Denoising+Backtranslation**

Our team attempted to reproduce the denoising and the denoising+backtranslation models  on the French-English language pair. We document our findings in the pdf here: https://github.com/anthonywchen/Unsupervised-NMT-Reproducibility/blob/master/ICLR_Reproducibiltiy.pdf

---

> ### Author Response · Authors · 2018-01-05
> **Re: Reproducibility of Denoising and Denoising+Backtranslation**
>
> Thanks for the effort put in reproducing our experiments. We would like to clarify that we did not observe any of the stability issues mentioned in the report. As usual with any deep learning experiment, there may be some subtle details that may have made it difficult to reproduce our results exactly. Moreover, it looks like the team missed some important details that were already present in the paper (e.g. they use fixed embeddings in the decoder, which we do not). In any case, we plan to release the entire package of code and scripts to reproduce our experiments once the submission has been accepted.

---

### Author Response · Authors · 2017-12-24
**New version of the paper available**

We would like to thank all reviewers for their detailed and insightful feedback. We have answered each specific point in the replies below, and uploaded a new version of the paper addressing them.

---

### Author Response · Authors · 2018-01-05
**New results**

We have uploaded a new version of the paper with the following new results, which aim to address the concerns raised in the reviews in relation to the semi-supervised experiments:

- We have tested the proposed method with only 10,000 parallel sentences, obtaining an improvement of 1-3 BLEU points over the unsupervised system. This reinforces that the proposed approach has potential interest beyond the strictly unsupervised scenario, showing that it can profit from a small parallel corpus that would be insufficient to train a conventional machine translation system.

- We have added new results for the comparable NMT system using the same parallel data as the semi-supervised systems. This was suggested by AnonReviewer1, and allows for an easier comparison between the semi-supervised and the supervised systems.

---

### Decision · Program_Chairs · 2018-01-29
**ICLR 2018 Conference Acceptance Decision**

**Decision:**

Accept (Poster)

**Comment:**

This work presents new results on unsupervised machine translation using a clever combination of techniques. In terms of originality, the reviewers find that the paper over-claims, and promises a breakthrough, which they do not feel is justified.
However there is "more than enough new content" and "preliminary" results on a new task. The experimental quality also has some issues, there is a lack of good qualitative analysis, and reviewers felt the claims about semi-supervised work had issues. Still the main number is a good start, and the authors are correct to note that there is another work with similarly promising results. Of the two works, the reviewers found the other more clearly written, and with better experimental analysis, noting that they both over claim in terms of novelty. The most promising aspect of the work, will likely be the significance of this task going forward, as there is now more interest in the use of multi-lingual embeddings and nmt as a benchmark task.